# Quantitative and Qualitative Distinctions between HIV-1 and SIV Reservoirs: Implications for HIV-1 Cure-Related Studies

**DOI:** 10.3390/v16040514

**Published:** 2024-03-27

**Authors:** Joseph C. Mudd

**Affiliations:** 1Tulane National Primate Research Center, Covington, LA 70433, USA; jmudd1@tulane.edu; 2Department of Microbiology and Immunology, Tulane University School of Medicine, New Orleans, LA 70112, USA

**Keywords:** NHP models of HIV-1, SIV reservoir, HIV-1 cure studies

## Abstract

The persistence of the latent viral reservoir is the main hurdle to curing HIV-1 infection. SIV infection of non-human primates (NHPs), namely Indian-origin rhesus macaques, is the most relevant and widely used animal model to evaluate therapies that seek to eradicate HIV-1. The utility of a model ultimately rests on how accurately it can recapitulate human disease, and while reservoirs in the NHP model behave quantitatively very similar to those of long-term suppressed persons with HIV-1 (PWH) in the most salient aspects, recent studies have uncovered key nuances at the clonotypic level that differentiate the two in qualitative terms. In this review, we will highlight differences relating to proviral intactness, clonotypic structure, and decay rate during ART between HIV-1 and SIV reservoirs and discuss the relevance of these distinctions in the interpretation of HIV-1 cure strategies. While these, to some degree, may reflect a unique biology of the virus or host, distinctions among the proviral landscape in SIV are likely to be shaped significantly by the condensed timeframe of NHP studies. ART is generally initiated earlier in the disease course, and animals are virologically suppressed for shorter periods before receiving interventions. Because these are experimental variables dictated by the investigator, we offer guidance on study design for cure-related studies performed in the NHP model. Finally, we highlight the case of GS-9620 (Vesatolimod), an antiviral TLR7 agonist tested in multiple independent pre-clinical studies in which virological outcomes may have been influenced by study-related variables.

## 1. Introduction

Antiretroviral therapy (ART) vastly improves the quality of life for persons with HIV-1 (PWH) but it is not curative, and viremia generally rebounds within weeks of interrupting ART [1,2]. The source of viral rebound is a pool of anatomically dispersed resting memory CD4 T cells that harbor latent but inducible HIV-1 DNA. Recent longitudinal studies indicate that after decades of ART, the reservoir of infectious HIV-1 does not decay and may even increase in size over long periods [2,3]. It has thus been a long-standing global health priority to pursue therapeutic approaches that either eradicate latent HIV-1 from the body or promote a state of long-term ART-free remission.

Nonhuman primate (NHP) hosts of progressive simian immunodeficiency virus (SIV) infection, namely rhesus macaques of Indian origin, represent the pre-eminent animal model of HIV-1 infection. Upon infection with SIV, animals display many salient pathological features of progressive HIV-1 disease, including but not limited to high viral loads [4], early gut mucosal CD4 T cell depletion [5], lymphoid tissue fibrosis [6], and immune cell exhaustion [7]. Like HIV-1, SIV also establishes a latent viral reservoir, and the adaptation of ART regimens that durably suppress viremia has allowed NHPs to be a valuable tool in the evaluation of HIV cure-related strategies [8]. NHP models of HIV-1 infection can accurately evaluate the clinical promise of cure strategies for several reasons. The first is that both latent HIV-1 and SIV respond in a very similar fashion to immune- or pharmacologic-based interventions [9,10,11,12,13]. Secondly, HIV cure approaches in NHPs can be easily evaluated against experimental ART treatment interruptions, which represent a ‘gold standard’ measure of the inducible replication-competent reservoir.

When HIV-1 cure strategies progress to clinical trials, success is evaluated by quantitative metrics. Specifically, whether a particular approach can decrease the overall size of the HIV-1 reservoir or delay HIV-1 rebound after treatment interruption. These endpoints measure HIV-1 reservoir behavior at the population level. Yet, it is important to point out that population-level behavior is manifested by the sum of the contributions of individual HIV-1 proviruses. Individual proviral clones are qualitatively heterogenous and differ by their degree or lack of lethal mutations, the propensity of the infected cell to undergo division, and the ability to which a provirus can be induced [14,15,16]. Thus, the qualitative properties of individual proviruses can underlie quantitative behavior at the bulk population level. Given that most HIV cure approaches rely on pre-clinical testing in NHPs, a relevant question to ask is how closely the reservoirs of SIV ART-suppressed NHPs resemble those of PWH. Data acquired through assays recently adapted to SIV have noted stark and potentially meaningful differences in the reservoir that overall may not be related to differing biology per se but to the shortened timeframe of NHP studies. In this review, we highlight properties of the SIV reservoir related to decay rate, intactness, and clonal structure and discuss how these are influenced by NHP study-related variables. When possible, we highlight how variations in these study designs may have impacted the outcome of HIV-1 cure interventions. For the purposes of this review, we restrict the survey of the literature in most instances to studies involving the SIVmac239 clone or SIVmac251 viral swarm, given that infection with highly passaged or genetically modified SIVs can vary widely in reservoir size and cell-type distribution [17,18]. Lastly, we offer guidance on study design with the goal of promoting rigorous interpretation of cure-related studies in the NHP model.

## 2. Timing of ART Initiation in SIV-Infected Macaques 

In untreated HIV-1 and SIV infections, the evolution of replicating viral strains is dynamic over time and reflective of adaptation to ongoing host immune pressures. It was originally thought that the latent reservoir is seeded predominantly by HIV-1/SIV strains circulating close to the period of ART initiation [19,20,21]. Recent evidence, however, suggests the latent pool of HIV-1 can also contain proviruses that are more archival, indicating that latency establishment and productive infection are concurrent throughout untreated infection [22,23,24]. The long-lived reservoir is thus influenced by the natural history of viral evolution during untreated infection, and the timing of ART initiation can significantly shape its size and genetic heterogeneity.

Initially, studies in macaques showed that SIV establishes a small yet generally incurable reservoir in gut mucosal and lymphoid tissues by day 3 post-infection, when plasma SIV is not yet detectable [25,26]. Further understanding of early reservoir formation has come from several clinical trials that enrolled participants diagnosed with HIV-1 very early in the disease course (RV217: NCT00796146) (RV254: NCT00796146) (FRESH cohort) [27,28,29]. The RV217 cohort longitudinally monitored the natural dynamics of HIV-1 DNA levels in the blood of high-risk participants from Thailand who acquired HIV-1 during Fiebig stages I/II. This study found that the size of the reservoir closely paralleled the known dynamics of plasma viremia, with exponential growth and peak HIV-1 DNA levels reached between 3 and 14 days after viral RNA detection (Fiebig I–III) [30,31] (Figure 1A). The set point of HIV-1 DNA was established between 4 and 6 weeks after viral RNA detection (Fiebig V–VI), where frequencies of infected cells are thought to remain relatively stable thereafter [32]. The period of Fiebig I–V also marks very dramatic changes in viral genome heterogeneity. Insights from the FRESH cohort have found that, during the period of rapid growth (Fiebig stages I/II), the genetic heterogeneity of viral DNA is limited but rapidly accumulates in Fiebig IV–V and continues to evolve into chronic untreated infection [33] (Figure 1B). The maturation states of CD4 T cells that harbor viral DNA through the early phases of acute infection differ as well. Insights from the RV254 clinical trial have found that during Fiebig I–II, blood CD4 T cells that are actively infected are predominantly of an effector memory phenotype and exhibit more restrictive T cell receptor diversity [23,30,34]. As early as Fiebig stage III, however, the central memory CD4 T cell compartment becomes increasingly susceptible to HIV-1 infection, and viral latency is established in more long-lived, polyclonal stem-like populations [30] (Figure 1B). In sum, studies from these cohorts of individuals who are infected very early suggest that the reservoir establishes a constant size by the set point of plasma viremia while remaining dynamic in both sequence diversity and cell-type distribution.

Unlike in clinical settings, the timing of ART initiation in NHP studies is a variable set by the investigator, and the issue of when to initiate treatment can profoundly impact the efficacy of approaches aimed at the eradication of the reservoir. Earlier ART treatment may favor the proof-of-principle of an intervention, whereas later treatment may favor its translatability. In general, interventions yielding an efficacious signal in clinical cohorts, most of which comprise subjects that initiated ART in chronic infection, are extremely limited and restricted to case reports [35]. Thus, while there is no overall standard for when to begin ART treatment in NHP studies, most studies have initiated ART within the first 8 weeks of SIV infection, favoring proof-of-principle of an approach over a design that is the most clinically relevant (Table 1).

There are several reasons why delaying the initiation of ART is likely to impede the ability of interventions to meaningfully impact viral persistence. The first is that as the disease progresses, the reservoir expands to include CD4 T cell maturation subtypes that are long-lived and likely more difficult to clear therapeutically. As stated earlier, the central memory CD4 T cell compartment becomes increasingly susceptible to HIV-1/SIV as the disease progresses [30,36,37]. By nature of their trafficking to lymphoid tissues, these cells come into frequent contact with γ-chain cytokines that sustain expression of BH3-only proteins known to counteract the death of infected cells by viral cytopathic effects or through immune-mediated clearance [38,39,40,41].

Secondly, as the disease progresses, the reservoir becomes more genetically diverse, and this could hamper eradication strategies that seek to engage innate or adaptive immunity. Studies have observed a high degree of CTL escape mutations and functional resistance to type I/II interferons among circulating viral strains seeded into the long-lived pool of replication-competent proviruses [42,43]. It is important to note, however, that despite this extensive diversity, the immune system continues to exert antiviral selective pressure on the reservoir [44,45]. Thus, while some proviral strains may be capable of evading host immunity if reactivated, this is likely not true for all HIV-1/SIV strains within the reservoir. Lastly, the extent of residual immune and tissue damage is certain to influence the outcome of HIV cure-directed therapies given that many agents under pre-clinical/clinical testing are predicated on engaging multiple facets of the immune system [13,46,47,48,49]. The majority of these induce transient innate and adaptive immune activation in the blood [13,48]. It is likely, however, that the bioactive impact of these agents is orchestrated within lymphoid tissues, and these organs are progressively damaged during untreated HIV-1 infection [6,50,51]. Several other facets of immunity that become progressively impaired and do not fully reverse with ART may also hinder the efficacy of cure-related approaches, including impaired antigen-presenting cell function, as well as the sustained expression of inhibitory immune receptors [49,52,53,54,55,56,57]. Thus, as infection progresses, the reservoir becomes more impervious to therapeutic intervention.

As stated, the majority of interventions tested pre-clinically have been conducted on NHPs, initiating ART between 1 and 8 weeks post-infection (Table 1). Within this time window, however, there is considerable variation, and comparing these studies can provide insights into how the variable of ART timing can influence the outcome of cure-related interventions. Very few cure approaches have been evaluated under small, less mature reservoirs that are associated with ART initiation less than 14 days post-infection [58,59,60,61,62]. A subset of these, however, were found to elicit a favorable impact and, in some instances, promote sustained ART-free remission. The first of these applied a ‘shock and kill’ strategy (i.e., reactivating latent HIV-1 and stimulating immune clearance) on SIV-infected macaques that initiated ART at 7 dpi [60]. At full virological suppression, these animals received the toll-like receptor 7 (TLR7) agonist Vesatolimod (GS-9620) alone or in combination with the anti-HIV broadly neutralizing antibody (bNab) PGT121. When ART was interrupted, six out of eleven animals in the combined Vesatolimod/PGT121 group exhibited delayed viral rebound, and in the remainder of these animals, viral rebound during the 196-day follow-up period was not observed [60]. The early ART notwithstanding, a caveat to this study was that it employed a strain of hybrid simian–human SIV (SHIV-SF162P3), many of which have variable replicative capacity and pathogenicity in vivo [63]. A more recent study employing wild-type SIVmac251 in neonatal macaques administered ART 7 dpi concurrently with a CCR5/CD3- or CD4-specific antibody had the goal of stimulating the clearance of target cells [59]. Upon interruption of ART, plasma SIV in three out of seven animals rebounded quickly, in two out of seven, rebound was delayed by 3 and 6 months, and in the remaining two, sustained remission was observed up to greater than 3 years post-ART cessation [59]. The remarkable outcomes of these two studies suggest two exciting possibilities. First, supplemental therapies could further promote the existing robust clearance of infected cells noted in subjects treated in Fiebig stages I/II of HIV-1 infection to meaningfully enhance the chance of remission [30]. Secondly, cure-related approaches tested previously on more mature reservoirs and deemed ineffective could be repurposed for extremely small reservoirs noted in very early ART. Nevertheless, the outcomes of these studies are largely not relevant to the majority of clinical scenarios. Most persons living with HIV-1 are diagnosed months to years after infection. Moreover, these studies were evaluated on reservoirs that were both quantitatively and qualitatively different than those of persons who are chronically infected, as are enrolled in most HIV-1 clinical trials [48,64,65,66,67]. In pre-clinical NHP studies that have initiated ART between Fiebig stages IV and VI (more closely approximating the reservoirs of PWH who are chronically infected), the success of interventions in these instances has been considerably more variable [61,68,69,70,71]. The vast majority of these have yielded either no significant changes to the frequency of cells harboring SIV DNA or have reduced replication-competent HIV-1 DNA to a level that elicited slight delays in viral recrudescence but not sustained ART-free remission (Table 1). To date, only one NHP study initiating ART at more clinically relevant periods has noted an intervention to induce durable ART-free remission in a small subset of animals. In this study, ART was initiated in macaques after 65 days post-SIV infection, approximating Fiebig V–VI, and after receiving serial doses of the TLR7-agonist GS-9620, durable ART-free remission was observed in two out of nine animals [10]. Interestingly, a parallel independent study employing the same intervention observed no delay in viral rebound in animals that initiated ART earlier, at 14 dpi. The reasons for this somewhat paradoxical outcome are unknown, but they suggest that the influence of ART timing on efficacy may depend on the particular nature of the intervention. Taken together, the timing of ART initiation can dramatically impact both the size and composition of the reservoir as well as the extent of immune and tissue damage. Each of these variables can, in turn, influence the efficacy of therapeutics aimed at targeting viral persistence. In the majority of pre-clinical studies, investigators have generally employed a study design that favors proof-of-concept (ART initiated 1–8 weeks post-SIV) vs one that is more clinically relevant (ART initiated in chronic SIV) (Table 1). However, when to precisely initiate ART in NHP studies must be a consideration unique to each study context.

**Table 1 viruses-16-00514-t001:** Summary of selected HIV-1 cure-related NHP studies denoting study variables, primary measurements, and outcome.

Study	Intervention	Class	SIV Inoculum	ART Initiation (dpi)	Duration of ART Pre-Intervention	Reservoir Measurement Assessed	Outcome
Lim et al. [10]	GS-9620 (Vesatolimod), GS-986	TLR7 agonist	SIVmac251	65	437	total SIV DNA, post-ART SIV rebound	SIV DNA reduction, SIV remission (2/13 animals)
Nixon et al. [11]	AZD5882	SMAC-mimetic	SIVmac239	60	420	inducible plasma viremia, SIV DNA	plasma viral blips, no reduction in SIV DNA
Policicchio et al [12]	Romidepsin	HDACi	SIVsmmFTq	65	250	inducible plasma viremia, SIV DNA	plasma viral blips, no reduction in SIV DNA
Borducchi et al [58]	PGT121, GS-9620	bnAb, TLR7 agonist	SHIV-SF162P3	7	672	total SIV DNA, post-ART SIV rebound	SIV DNA reduction, SIV remission (5/11 animals)
Deere et al. [60]	CD4 nAb, CCR5 nAb	biologic	SIVmac251	7	16	post-ART SIV rebound	SIV remission (2/7)
Varco-Merth et al [61]	Rapamycin	mTOR inhibitor	SIVmac239	12	230	total SIV DNA, post-ART SIV rebound	no impact
Gramatica et al [62]	Tideglusib	Akt/mTOR agonist	SIVmac239X	12	672	total SIV DNA	no impact
Pino et al. [68]	FTY720 (fingolimod)	S1P-receptor antagonist	SIVmac239	42	190	total SIV DNA	no impact
Swainson et al [69]	anti-IFNa	biologic	SIVmac251	48	84	total SIV DNA	faster decline
Micci et al [70]	IL-21	cytokine	SIVmac239	60	210	total SIV DNA, post-ART SIV rebound	SIV DNA reduction, no delay in SIV rebound
Harper et al. [71]	anti-IL-10	biologic	SIVmac239	35	210	total SIV DNA	SIV DNA reduction
Dashti et al [56]	AZD5882 + SIV nAb	SMAC-mimetic, biologic	SIVmac239	56	609	viral outgrowth (QVOA)	reduced replication-competent SIV DNA
Harper et al. [47]	anti-CTLA4, anti-PD-1	biologic	SIVmac239	60	407	intact proviral DNA, post-ART SIV rebound	intact SIV DNA reduction, no delay in SIV rebound
Del Prete et al [58]	GS-9620 (Vesatolimod)	TLR7 agonist	SIVmac239X	12	450	intact proviral DNA, post-ART SIV rebound	no impact

## 3. Decay Kinetics following ART Initiation in NHPs

The reservoir undergoes several phases of decay when new infections are blocked with ART. The early phases of this decay are particularly dynamic. Given that pre-clinical testing of agents in the NHP model is oftentimes evaluated during this period, it is critical to understand how the SIV reservoir decays naturally with ART so that it can be differentiated from interventional decay. It is important to note that, during untreated infection and shortly after ART initiation, a significant amount of viral DNA is not integrated into the host genome, and exists as labile linear viral DNA (t_1/2_ = 2d), 1- and 2-LTR circles, which exhibit greater stability but are not copied or inherited during cell division [72,73,74]. Moreover, only a fraction of the integrated viral DNA is intact and considered replication-competent (see below). It is thus more accurate during this period to refer to the term total viral DNA rather than proviruses, the latter of which represent most viral DNA measured during long-term ART suppression. The most detailed study to longitudinally track the dynamics of intact SIV DNA following ART initiation was performed by Fray et al. [24]. Animals were infected with SIV_mac251_ and left untreated for ~1 year upon initiating a standardized ART regimen, with rapid suppression by 4 weeks of treatment. During 4 years of continuous therapy, the authors observed a dynamic decay of intact SIV DNA, which consisted of three distinct phases [24]. The initial 4 weeks of ART induced a rapid decay of intact SIV DNA (mean t_1/2_ = 3.3 days). This was followed by a slower, second phase associated with the decay of intact SIV DNA harbored in longer-lived cells (mean t_1/2_ = 8.1 months). After a mean of 2.3 years of ART, a third phase became evident that was associated with virtually no decay. As in PWH, defective SIV genomes differed markedly from intact proviral DNA in their decay kinetics and did not follow a uniform pattern [24], likely reflecting the influence of immune selection pressures exerted on the latter [44,45,75,76,77].

When compared to SIV DNA decay dynamics in the study noted by Fray et al. [24], ART initiation in PWH induces a similar multiphasic decay of HIV-1 DNA, although particular differences were noted (Figure 2A). First, the decay of viral DNA close to ART initiation is more rapid in NHP (mean t_1/2_ 3.3 d vs. 12.9 d) [24,78]. The clearance of intact SIV DNA is also more rapid in the second phase (mean t_1/2_ 8 months vs. 19 months), whereas the decay of intact viral DNA that constitutes the stable reservoir during the third phase is negligible in both PWH and SIV_mac251_-infected animals [24,78]. It remains to be determined whether the ART-mediated decay rates observed in the study performed by Fray et al. are generalizable to all NHP studies. The above analysis was performed on a group of 10 animals, which is relatively small compared to clinical studies assessing reservoir decay rates in large cohorts of PWH initiating ART [79]. ART was also initiated ~1 year after infection, much later in the disease course than most cure-related NHP studies (Table 1). While CD4 T cell counts at the time of ART initiation were not reported in the study by Fray et al. [24], it is likely that nadir CD4 T cell counts were much lower in these animals, which is important given that nadir CD4 is highly predictive of reservoir size in long-term suppressed PWH [80,81,82,83].

Importantly, it is now clear that HIV-1 reservoir dynamics enter a fourth phase in decades-long suppressed PWH that is characterized by an apparent slow increase in reservoir size, a dynamic that was not captured in 4 years of continuous ART in the NHP study by Fray et al. (Figure 2A) [2,24]. The inflection point between the third and fourth phases likely underlies a shift in equilibrium between the opposing forces of cellular proliferation and selection forces that act on cells harboring intact, inducible proviruses. Cellular proliferation replenishes the reservoir whereas immune selection eliminates intact proviral DNA that is transcriptionally active. Over decades of therapy, what remains is a pool of CD4 T cells continually replenished by physiologic proliferation that harbor deeply latent proviruses integrated within repressive chromatin regions of the genome [45]. While not as comprehensively described, these opposing forces are likely to be operable in ART-suppressed SIV+ NHPs as well. Thus, it is conceivable that over a period longer than 4 years of continuous ART, reservoir dynamics in NHPs may also transition to an analogous fourth phase of very slow reservoir expansion. The precise time of this inflection may differ as the lifespan of NHPs in captivity is roughly one-third that of humans; however, determination is difficult given the cost of maintaining animals under study for prolonged periods.

It is important to note that cure studies in NHPs are typically conducted during the more rapid second phase of decay (≤2 years ART), whereas most subjects enrolled in clinical trials have been on ART much longer and exhibit more stable reservoirs (Figure 2A) [48,64,85,86,87]. The differing time windows of intervention may have several implications for cure-related studies. The first is that the natural decay of the reservoir in NHP models is likely to be more robust across an interventional window when compared to clinical settings. Thus, natural decay could more easily be mistaken for therapeutic decay. To illustrate this point, a study by Kumar et al. evaluated the impact of toll-like receptor agonists on intact viral DNA before and after a 22-week therapeutic window in a small cohort of SHIV-infected macaques, initiating the intervention at 64 weeks post-ART initiation [84]. A slight decreasing trend of intact SHIV DNA was noted between pre- and post-intervention [84]. When these measurements were corrected for SHIV DNA decay rates that were predictive of having occurred naturally with ART alone, trends observed within the interventional group were abolished [84]. In cure-related studies, it is thus critical to account for the natural decay of the reservoir by employing robust no-intervention control groups that are evaluated at parallel timepoints to the groups receiving the intervention.

In addition, because interventions are started earlier in NHP models, a greater fraction of SIV DNA will exist as unintegrated 2-LTR circles that are not part of the stable reservoir (Figure 2B). These forms of viral DNA are still actively decaying during typical interventional windows in studies employing NHP models, and at 1 year of ART, they remain approximately 10-fold higher than levels of 2-LTR circles in long-term suppressed PWH [88]. Forms of unintegrated viral DNA could thus disproportionately impact decay rates if interventions are employed at time periods ≤2 years of ART, and accounting for these in the NHP model is important.

## 4. Proviral Intactness of the SIV Reservoir

Given the widespread use of SIV-infected NHPs for cure studies, it is relevant to ask how closely their proviral landscapes resemble those of long-term suppressed PWH. A major feature of persisting proviral DNA in treated individuals is that most viral genomes harbor lethal mutations that render them incapable of replicating if ART is interrupted. These include large internal deletions that arise from template switching during reverse transcription, APOBEC3-mediated hypermutations, point mutations, and packaging signal deletions. Proviruses that lack any of these defects are classified as intact, and by near-full length (NFL) viral DNA sequencing, they comprise only 2–10% of the persisting HIV-1 DNA in the blood of most ART-suppressed PWH [89,90,91]. Two studies have employed near-full-length (NFL) sequencing to interrogate the intactness of persisting SIV proviruses [88,92]. Both of these studies noted that deletions were the most common type of defect [88,92], similar to that of HIV-1. A distinguishing feature noted in the study by Bender et al. was that a higher fraction of defective SIV sequences exhibited APOBEC3-mediated hypermutation, resulting in fatal G → A conversions that led to altered start and stop codons [88]. While mechanistic insights into this distinction are currently unknown, the observation may reflect (1) differences in the expression or activity of APOBEC3 family members in NHPs, (2) differences in *vif*-induced APOBEC3 degradation between HIV-1 and SIV, or (3) the shorter timescale of ART [93,94,95].

The predominance of replication-defective proviral sequences presents a challenge for both accurately measuring the reservoir and evaluating viral eradication strategies. Initially, HIV-1 and SIV reservoirs were quantified by amplifying a small, conserved region of the HIV-1/SIV *gag* open reading frame. This method, while straightforward and reliable, cannot discriminate between intact and defective proviral DNA and thus heavily overestimates reservoir size. NFL sequencing is the gold standard in determining proviral intactness, yet this technique is not suitable for quantitation because experimental inefficiencies often preclude adequate sampling depth and the technique biases for the amplification of shorter, defective proviral DNA that contains large internal deletions [96]. Quantitation of replication-competent HIV-1/SIV reservoirs is further complicated by the fact that not all intact proviruses are readily induced ex vivo in viral outgrowth assays [96]. To overcome these limitations, digital droplet-based multi-plex PCR assays were developed that include the intact proviral DNA assay (IPDA), quadruplex quantitative PCR (Q4PCR), and five-target IPDA (5T-IPDA) [97,98,99]. An advancement of these assays is that they allow for the discrimination of intact and defective proviruses in a scalable, labor- and cost-effective manner. The IPDA is now validated for both SIV_mac239_ and SIV_mac251_, as well as SHIV viral strains [47,84,88], and an initial, surprising observation uncovered by the IPDA and NFL sequencing was that, relative to HIV-1, the SIV reservoir comprised a significantly greater fraction of intact and presumably functional proviral DNA (Figure 3). In the study by Bender et al., intact proviral sequences accounted for roughly 30% of persisting SIV proviruses [88]. It is important to note that this study employed animals that initiated ART roughly 2 years after infection, significantly later than most NHP studies. Other NHP studies assessing proviral intactness have initiated ART sooner. In one of these, ART was initiated 1 year after SIV_mac239_ infection, and these authors observed by NFL sequencing that ~50% of SIV DNA was classified as fully intact after 1 year of therapy [92]. The same group of investigators studied an additional set of animals initiating ART 1 month after infection [26], a time window similar to that employed in many pre-clinical HIV-1 cure studies (Table 1). Here, an even greater fraction of proviral sequences in blood were classified as intact, constituting >80% of persisting SIV genomes [26]. These three studies collectively illustrate that the longer infection is allowed to progress untreated, the more defective proviral DNA accumulates, and suggest that the species-specific difference in reservoir intactness between clinical and NHP models is not due to the underlying biology per se but rather differences in the timing of ART initiation. Corroborating this evidence is an observational study that assessed proviral sequence diversity in two individuals with subtype-C HIV-1 initiating ART at Fiebig stage II, where the majority of viral DNA genomes persisting in the reservoir were found to be fully intact [33].

Higher fractions of intact proviruses that predominate the reservoir in SIV may have several implications for the interpretation of viral eradication strategies. The first is that interventions deemed successful in NHPs may have a high probability of exhibiting efficacy in clinical trials, given that the bar for obtaining remission in SIV-infected NHPs may be higher due to greater amounts of replication-competent proviral DNA. The second is that evaluating the efficacy of cure approaches in the NHP model may be more straightforward. If ART is initiated within the early months of infection, most of the reservoir will likely be intact. The efficacy of an intervention measured by total SIV DNA may thus accurately approximate changes to the intact and presumably functional reservoir measured by the SIV IPDA. It is important to note that, in clinical settings, this is notably not the case. There is at least one instance in which the impact of a cure intervention was measurable through intact but not total HIV-1 DNA [13]. Employing the IPDA in clinical settings is also more nuanced given the profound genetic heterogeneity that can exist in the reservoirs of PWH. Polymorphisms within the IPDA primer/probe binding sites can cause assay failures and lead to underestimation of intact proviral sequences [16,100,101]. The IPDA may also misclassify some truly defective proviruses as intact. These instances arise from HIV-1 DNA that is intact at both assay primer/probe locations but harbors lethal mutations outside of these sites [102].

This is not to say, however, that the use of the IPDA in SIV does not come without its own unique set of challenges. The SIV reservoir harbors significantly higher fractions of hypermutated proviruses. It is not uncommon for proviral DNA to harbor G → A mutations at both primer/probe locations of the IPDA, leading to a resultant no signal and the misclassification of droplets containing defective proviruses as uninfected droplets (Figure 3). Because of this, particularly in SIV, the IPDA is not suitable for quantifying total proviral DNA and likely overestimates the proportion of total proviruses that are intact (Figure 3). For these purposes, digital-droplet reactions targeting SIV *gag* that more accurately quantify total proviral DNA should be conducted in parallel with the IPDA. Alternatively, the IPDA can be modified with primer/probes that hybridize specifically to hypermutated proviral sequences [103].

## 5. Clonotypic Structure of the SIV Reservoir

The bulk dynamics of the HIV-1/SIV reservoir are driven by the cumulative behaviors of individual proviral clones. When assessing proviral clones in long-term suppressed PWH by single genome sequencing of the HIV-1 *env* gene, an early observation was that oftentimes many of them were identical [104]. These proviral clones, soon confirmed by other groups, were found to comprise full-length replication-competent HIV-1 genomes [15], persist longitudinally at year-long intervals [105], and were capable of producing infectious viruses ex vivo and in vivo [106,107,108]. The fact that HIV-1 proviral clones also share identical host genomic integration sites indicates that clonal sequences are the product of cellular proliferation. Antigen-driven proliferation, in particular, that is driven by recurring antigens, enriches the reservoir in select CD4 T cell clonotypes over time [106,109]. Clonal skewing of the reservoir increases progressively with the amount of time on ART [45,110,111,112], and long-term suppressed PWH who exhibit negligible decay of the reservoir over time are characterized by proviral landscapes that are highly clonally skewed [45]. Thus, the significance of clonal expansion is that it continually replenishes the reservoir.

Two studies have assessed the clonal structure of the reservoir in SIV-infected NHPs, and both observed identical proviral sequences that were similar to those of HIV-1. Yet, relative to long-term suppressed PWH, the proviral landscape in ART-suppressed NHPs was comprised of significantly fewer clonally expanded populations (Figure 2B) [88,113]. The first of these studies sampled over 400 intact *env* amplicons or near-full-length proviral clones from blood CD4 T cells of seven chronic-SIV_mac251_-infected macaques on daily ART for 9 months. Strikingly, only two pairs of clones were found to be identical from a single animal, constituting 1.7% of the sampled proviruses [88]. The authors found a similar paucity (three pairs, one triplicate) of identical clones in chronic SIV_mac239X_-infected macaques sampled after 53–58 weeks of ART [88]. This study employed animals that were well-progressed prior to initiating ART and assessed clonality by the SIV genetic sequence, which may not rule out independent infections by the same SIV variant. A second study assessed clonality by proviral integration site in blood and lymphoid tissues among SIVmac239-infected macaques that initiated ART 1-month post-infection, a timeframe similar to that of many cure-related NHP studies. Following 2 years of daily ART, in 150 total clones, the mean fractions with identical SIV integration sites observed were 4.8% and 8.7% in blood and lymphoid tissues, respectively [113]. These are significantly less than the clonal frequencies observed in long-term suppressed PWH, which oftentimes comprise 40–60% of proviruses sampled in blood [45,106,107,114,115], and suggest that the NHP model may not entirely reflect the proviral landscapes of PWH treated with long-term ART.

What could explain the relative paucity of clonal sequences in reservoirs of SIV-infected NHPs? It is unlikely that this underlies a unique biology in NHPs, as homeostatic processes that shape the reservoir are governed by parallel mechanisms in both humans and NHPs [116]. The immune systems of both species are also exposed to similar recurring antigens in the form of persistent β-herpesviral infections that drive effector memory T cell expansion, such as those derived from cytomegalovirus (CMV) [117,118]. Differing proviral landscapes may be more likely to be shaped by the condensed time window of NHP studies (Figure 2B). Clonal skewing of the reservoir is the additive effect of both negative selection by immune pressure and positive selection by antigen stimulation. The imprint of these two processes is observed over years, if not decades, in the reservoirs of PWH as opposed to those of NHP models that typically employ treatment windows of 1–2 years. If the duration of ART was prolonged in NHP studies, it is possible that some proviral clones would expand to frequencies on par with those observed in PWH. The longest study to date sampled single SIV proviral clones longitudinally in some animals for up to 4 years of daily ART [24]. As in PWH, the frequency of identical sequences increased progressively with time on ART, with identical sequences in the reservoir comprising a mean of ~10% of sampled proviruses after 1 year of therapy and a mean of 20% when ART was continued for up to 4 years, with some animals exhibiting clonal frequencies as high as 40% [24].

It remains to be determined whether lower frequencies of clonal sequences in SIV are shaped in any way by inherent biological differences of the virus or host. Viral loads at the time of ART initiation in untreated SIV infection can oftentimes exceed those of HIV-1 by roughly 1 log [47,66,88]. A paucity of clonal sequences in treated SIV infection could reflect, in part, viral dissemination in a more diverse array of CD4 T cell clonotypes or higher contributions of polyclonal naïve CD4 T cells to the infected cell pool [119,120]. Regardless of the underlying nature, the studies above suggest that cure-related approaches in the NHP model are evaluated on proviral landscapes that are markedly less clonally skewed than those of long-term suppressed PWH. Other experimentally studied species of NHPs may recapitulate the proviral landscapes of PWH more closely. For example, pigtailed macaques are an additional yet less frequently employed NHP model of HIV-1. An underappreciated aspect of these species is that they exhibit significantly higher proportions of effector memory CD4 T cells across tissues when compared to those of rhesus macaques [121], and these species may better model clonal expansion of the reservoir.

Lastly, one outstanding question is whether distinctions within the proviral landscapes between NHPs and humans lead to differences in reservoir behaviors at the quantitative level. Although SIV reservoirs in ART-suppressed NHPs are generally less clonal, particular studies unique to NHPs have the potential to shed light on the contribution of clonal expansion to bulk reservoir size. In this regard, specific pathogen-free rhesus colonies maintained by several of the national primate research centers are seronegative for persistent herpesviruses such as CMV throughout life and exhibit a striking paucity of clonally expanded effector memory CD4 T cells in circulation, even into late adulthood. Comparative studies that evaluate reservoir dynamics in the presence (CMV-seropositive) or near absence (CMV-seronegative) of clonal proliferation could define how this mechanism independently contributes to SIV persistence over proliferation induced by other stimuli.

## 6. Vesatolimod: A Case Study on the Impact of NHP Study-Related Variables on HIV-1 Cure-Related Outcomes

To illustrate how study-related variables, including viral inoculum, time to ART initiation, and duration of ART before intervention, may have shaped the outcome of HIV-1 cure endpoints in NHP studies, we present the case of GS-9620 (Vesatolimod), a TLR7 agonist with antiviral properties tested in multiple clinical and pre-clinical settings. HIV-1 cure agents have broadly relied on two interdependent modes of action to facilitate reservoir clearance: (1) reactivating latent SIV to allow infected cells to become visible to the immune system, and (2) enhancing innate and adaptive immunity to eliminate these cells. Compounds tested in early clinical trials exhibited the ability to reactivate latent HIV-1 but did not effectively stimulate immune effector function and, in some cases, were associated with immunosuppressive properties [86,122,123,124,125]. GS-9620 was one of the first single-agent compounds discovered to promote both modes of action in vitro [126]. Based on these antiviral properties, a study employing 21 adult male rhesus macaques was performed by Lim et al. to assess the in vivo impact of GS-9620 on SIV reservoir clearance [10]. Macaques were infected intrarectally with SIV_mac251_ and began receiving daily ART 65 days after infection. ART was continued for approximately 400 days, and the animals were subsequently administered 10 doses of oral GS-9620 (0.05 or 0.15 mg/kg) every 2 weeks, followed by a resting period of 3 months while maintaining ART. After the treatment pause, GS-9620 was resumed for an additional nine doses at the same concentration and frequency. Concurrently, a separate independent study employing GS-9620 was performed on six male adult rhesus macaques by Del Prete et al. [58]. This study employed an identical treatment regimen of two bi-weekly GS-9620 administration windows separated by a prolonged resting phase but differed by several notable study parameters to that of Lim et al.: (1) animals were inoculated with a barcoded SIV clone SIV_mac239X_ instead of a SIV_mac251_ swarm, (2) ART was initiated earlier, at 14 days post-infection, (3) investigators waited longer to initiate GS-9620 treatment (550 vs. 400 days on ART), and (4) a portion of the administrations were given at a higher dose (0.5 vs. 0.15 mg/kg). Both studies observed GS-9620 to be bioactive, eliciting transient but robust induction of innate and adaptive immunity [10,58]. Only the study by Lim et al., however, noted a virological impact. GS-9620 administration in these animals exhibited transient increases in plasma viremia, significant reductions in levels of total and inducible SIV proviruses, and, interestingly, long-term remission in two out of nine animals when ART was discontinued. Thus, despite a comparable bioactive response to GS-9620, SIV reactivation and subsequent clearance of latent infected cells were only noted in one of these studies. A lack of virological impact with GS-9620 alone was also noted in separate independent pre-clinical studies in SIV_mac251_ or SHIV-infected animals [60,127].

A relevant question to ask is whether study design differences contributed to the diverging impact of GS-9620 on virological outcomes. It is possible that differences in the inoculums of SIVmac239X (viral clone) versus SIVmac251 (viral swarm) may have contributed, although the frequencies of total viral DNA at the baseline of the two studies were comparable [10,58]. Other study-related variables influencing the virological impact of GS-9620 may be related to the timing of ART initiation or the duration of ART prior to administering the compound. These in turn could impact both the qualitative aspects of the reservoir and the degree of immune and tissue damage sustained during untreated SIV infection. While reservoirs in either study were not assessed for intactness or clonality at the single-genome level, in clinical trials, GS-9620 has been administered in cohorts with known distinctions among proviral landscapes and differing degrees of immune dysfunction, and these studies were associated with diverging virological responses to GS-9620. A benefit of these studies is that all study parameters between the two were nearly identical, with the important exception of cohort composition: one study enrolled typical HIV-1 progressors, and the other study enrolled HIV-1 controllers (HIV-1 RNA, 50 to 5000 copies/mL at ART initiation). HIV-1 control is most consistently associated with immunity, which is thought to be superior to that of typical progressors [128,129,130]. Moreover, while HIV-1 controllers as a group are heterogenous, reservoirs in subjects that control HIV-1 below detection limits (i.e., elite controllers) are significantly smaller and are qualitatively different at the single clone level than those of typical progressors [131]. In the cohort of HIV-1 controllers, GS-9620 was associated with the induction of immune activation, reduced frequencies of intact proviral HIV-1 DNA, and modest but significant delays in viral rebound when ART was discontinued. In contrast, no significant change in HIV-1 reservoir size was seen in the cohort of typical progressors on ART in spite of similar immune induction with GS-9620 [64]. The two studies collectively suggest that host and virological characteristics may significantly influence the outcome measurements of HIV-1 cure studies. In all studies that have employed GS-9620, the general consensus on the antiviral action of the drug is that it alone did not mediate a consistent impact on viral reservoirs, although it consistently exhibits a transient robust stimulation of innate and adaptive immunity. Thus, employing GS-9620 in combination with other agents remains a promising and viable treatment strategy.

## 7. Concluding Remarks

The utility of NHPs as pre-clinical models in HIV-1 cure studies ultimately depends on how closely they can inform safety and efficacy measures in clinical trials. In this respect, they are unmatched when compared to other animal models of HIV-1 infection. It is important to note, however, that reservoirs in NHPs generally differ in important qualitative respects when compared to those of PWH. In general, cure-related studies in NHPs are conducted on reservoirs that are more genetically intact, less clonally skewed, and exhibit higher rates of natural decay during typical intervention windows (Figure 2 and Figure 3). While the degree to which species-inherent virus/host factors contribute to these distinctions is currently unknown, they are likely to be shaped significantly by the condensed timeframe of NHP studies. ART is typically initiated sooner, and animals are virologically suppressed for shorter amounts of time before receiving interventions. Because these variables can impact both quantitative and qualitative aspects of the reservoir and are dictated experimentally by the NHP investigator, we offer the following recommendations in study design: (1) While there is no general standard for when to precisely initiate ART in NHP studies, the timing of ART should be considered within the context of each particular study. For those seeking to establish the proof-of-principle of an approach, it is more beneficial to initiate ART between Fiebig II and VI (1–8 weeks post-infection) but prior to chronic infection. On the other hand, for studies seeking to model the translatability of a particular approach, it is more relevant to initiate ART in individuals with chronic SIV infection. If an efficacious signal is observed in this setting, these therapies may be the most clinically relevant given that persistent reservoirs formed with ART initiation during chronic infection are likely more difficult to clear therapeutically. (2) Especially if study timelines cannot accommodate ≥2 years of full virologic suppression prior to intervention, careful, no-treatment control groups should be employed and evaluated at parallel timepoints of treatment groups to distinguish natural versus interventional decay across a given time window.

Lastly, we highlight several unanswered questions relating to HIV-1 cure therapies, with particular relevance to the NHP model. First, it is currently unknown whether viral DNA harbored in other cell types, such as myeloid cell populations, impacts reservoir dynamics and whether they are equally sensitive to therapeutic modalities. In NHP studies, it is not uncommon to quantify viral DNA by whole PBMC or whole tissue rather than purified CD4 T cells. While the contribution of myeloid populations to the replication-competent reservoir remains controversial [132,133,134], persisting viral DNA can be detected in these cell types of both PWH and NHPs and may be capable of viral outgrowth [135,136,137,138,139,140,141].

Finally, an important but unanswered question remains: how do the noted qualitative differences in reservoirs of NHPs relate precisely to quantitative metrics evaluated in cure studies? A clinical study of PWH who were ART suppressed for over 2 decades found that reservoirs of subjects who maintained viral control after discontinuation of therapy exhibited robust signatures of immune selection, being dominated by large proviral clones located in transcriptionally silent regions of the host genome [45]. This may suggest that, as a result of immune selection, the inducibility of the reservoir decreases over time, and given that NHP studies are carried out under condensed timeframes, pre-clinical testing of cure therapies may be performed under reservoirs that are more inducible than those of long-term suppressed PWH. It is important to point out, however, that viral outgrowth as a result of treatment interruption (above study) is inherently different from inducibility mediated by an exogenous stimulatory agent. Interestingly, a recent study by Mcmyn and colleagues found that the ex vivo inducibility of the reservoir in PWH does not decay between a mean of 6 and 22 years of ART and may even increase as a result of infected cell proliferation [2]. This study, however, did not evaluate inducibility during the first 1–2 years of ART, a window of time often evaluated in pre-clinical testing of cure-related therapies. Thus, a relatively straightforward but clinically meaningful experiment would be to evaluate the inducibility of the reservoir at earlier timepoints that are associated with more rapid decay rates of intact proviral DNA. These questions notwithstanding, it is undoubtedly certain that NHPs will continue to be the most accurate pre-clinical model to inform the safety and efficacy of future HIV-1 cure-related therapies to be tested in clinics.

## Figures and Tables

**Figure 1 viruses-16-00514-f001:**
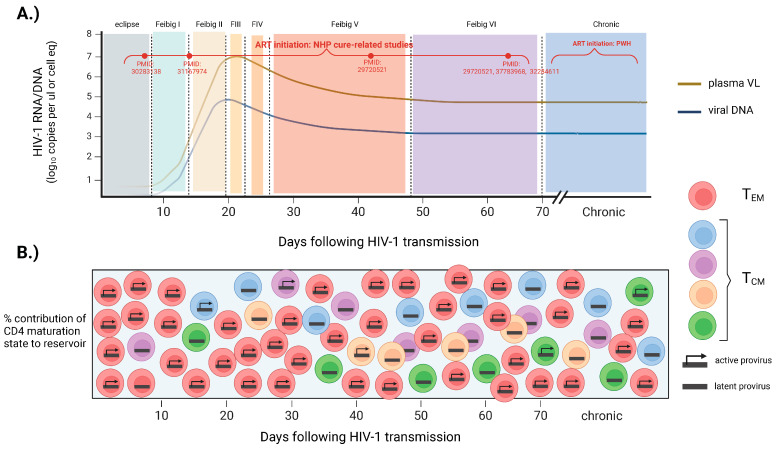
Typical periods of ART initiation in NHP studies. (**A**) Dynamics of plasma VL and viral DNA across the 6 Fiebig stages of acute HIV-1 infection. The period in disease course at which ART was initiated in several NHP studies employing HIV cure therapeutics is noted in red. In most clinical scenarios, HIV-1 is diagnosed as chronic infection. (**B**) The contribution of CD4 T cell maturation states harboring the reservoir across different Fiebig stages of acute HIV-1/SIV and at differing periods of ART initiation in published NHP cure-related studies.

**Figure 2 viruses-16-00514-f002:**
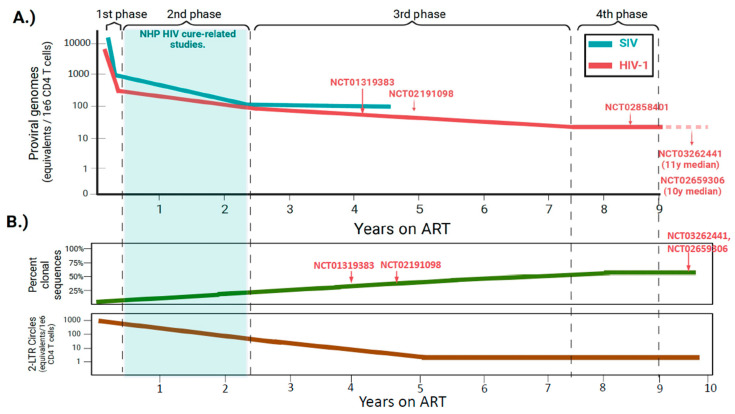
ART-induced viral DNA decay rate, proviral clonotypic landscape, quantities of unintegrated viral DNA, and typical intervention windows are major factors that differentiate NHP pre-clinical from clinical studies. (**A**) Decay rates of viral DNA in SIV NHP (blue) and HIV-1 (red) after initiating ART. Distinct phases of viral genome decay are noted, with the majority of interventions tested in the NHP model of HIV implemented significantly earlier after virologic suppression than those of clinical trials (examples noted in red by the “NCT” clinical trial number). (**B**) Longitudinal dynamics of the clonal landscape of integrated viral DNA and quantities of unintegrated forms of viral DNA that exist as 2-LTR circles. In general, the majority of pre-clinical NHP studies are performed on reservoirs that are less clonotypically skewed and comprise higher levels of unintegrated viral DNA compared to reservoirs of participants enrolled in clinical trials. This illustration has been adapted from Kumar et al and the concepts of this illustration are derived from the work of this study [84].

**Figure 3 viruses-16-00514-f003:**
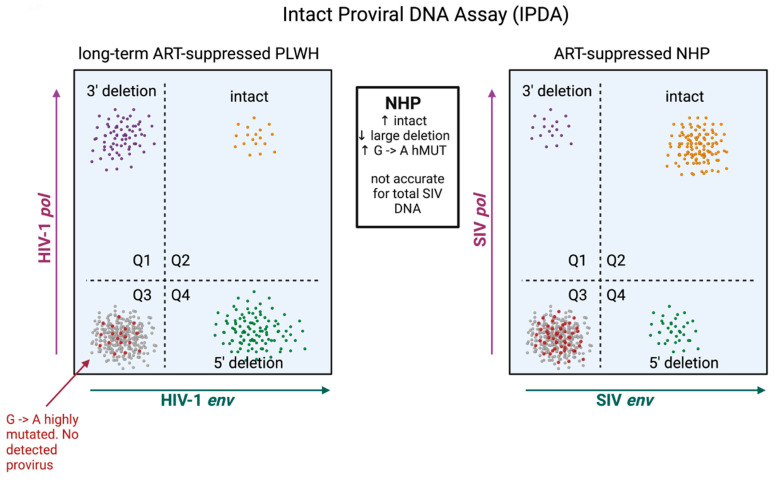
Distinctions in genomic integrity between HIV-1 and SIV reservoirs measured by the intact proviral DNA assay (IPDA). Representative differences in proviral landscapes of long-term suppressed PLWH and ART-suppressed NHPs illustrated by the intact proviral DNA assay. Major distinctions in NHP include higher frequencies of intact viral genomes, lower frequencies of viral genomes harboring large deletions, and higher frequencies of highly mutated viral genomes that cannot be detected by conventional IPDA probes. This illustration has been adapted from Bender et al. and the concepts are derived from the work of this study [88].

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
