# Peer review of "Quantitative and Qualitative Distinctions between HIV-1 and SIV Reservoirs: Implications for HIV-1 Cure-Related Studies"

_viruses, 2024, doi:10.3390/v16040514_

Round 1

Reviewer 1 Report

Comments and Suggestions for Authors

This is a timely review article covering an important topic in HIV cure research. As reservoir characterization has become more sophisticated, a number perceived differences between HIV reservoirs in humans and reservoirs established in SIV based NHP models have emerged. The author does a nice job of identifying these key distinctions, and then taking a nuanced view of why these differences might exist and what they might mean for the use of NHP models for study persistent HIV reservoirs and evaluating approaches to control or eliminate them, pulling from the appropriate relevant literature.  The review structure is well-organized.

There are a few areas for improvement that I think would elevate the utility and impact of this review

Major Points

1.       The weakest section of the manuscript is the “Timing of ART initiation in SIV-infected macaques” section. The author endeavors to make some generalizable points about the potential impact of ART timing on reservoir and cure studies, but many of the points seem to ultimately contradict one another and the message gets quite muddled. Initially the author spends some time nicely laying out the dynamic shift in cellular phenotypes harboring viral genomes that occurs during acute infection, identifying “0-8 weeks” (line 90) as the time window when this evolution in reservoir cell type occurs. The author then notes that a potential shortcoming of many SIV studies is the common use of ART initiation during this dynamic time period. However, the author then states that one of the justifications used by SIV researchers start ART during these early time periods is because “Initiating ART shortly after reservoir set point thus allows HIV cure approaches to be evaluated on fully matured reservoirs” (line 97). It is thus unclear when the author suggests reservoirs are still dynamically changing vs fully mature.  This point seems to suggest that even though SIV researchers are starting ART early, they are still actually past the point when the reservoir cellular phenotypes are dynamically changing, which runs counter to the original point the author was making.  This section could use some revision for clarity, as the point gets lost or muddled.

2.       Related to point 1, line 105 states that “Very few cure approaches have been evaluated under small, less mature reservoirs that are associate with very early ART.” The author then seems to try to make a broad point thdoes not define what “very early ART” means, but the prior passages would suggest that the author means starting ART <8 weeks post infection.  In the next sentence, the author states “Ones that have however were found to elicit favorable impact and in some instances promoted sustained ART-free remission.”  However, Table 1 shows a number of studies where most initiated ART during this early period when the cellular phenotypic composition of the reservoir is still changing dynamically, but the outcomes were a mixed bag with most studies showing no evident impact on the reservoir.  Even if the author is actually referring to ART initiated during an even earlier window than this, say <2 weeks post-infection, the suggestion that earlier ART increases the likelihood of a promising result doesn’t seem consistent with the field as a whole. The vestolimod example given in the paper itself actually seems to make the counterpoint, where a study involving earlier ART showed no effect but a study involving later ART showed a dramatic effect.  I think a much safer and accurate position would be that the timing of ART initiation can impact reservoir composition, immune responses, and numerous other factors which may impact the activity (or the ability to measure activity) in ways that are intervention-specific.

3.       Line 132: “To date, therapies tested on fully-formed reservoirs have yet to reduce the replication competent viral DNA to a degree that results in a meaningful delay in viral rebound.” Related to the above 2 points, the Lim et al vesatolimod study would seem to run counter to this point.

4.       Much is made about the potential impact of unintegrated forms of DNA, such as 2-LTR circles, on viral DNA measurements based on the fact that the levels of unintegrated DNA are higher in SIV infected macaques on ART that in humans on ART with HIV. I think this point is a bit overblown.  Although it is true that the relative levels are higher in monkeys, they still represent a very small fraction of the total DNA population during ART, so potential changes in this pool seem unlikely to meaningfully impact total DNA levels in a way that would lead to data misinterpretation.

5.       Because this review article is focused the nuanced issues surrounding comparisons between reservoir assessments in SIV infected macaques and humans with HIV, I think that the article would benefit from a bit more nuanced discussion of the Fray et al study, rather than viewing the Fray et al numbers as necessarily indicative of decay rates across SIV/NHP studies.  In particular, while reservoir decay rates in humans have typically been derived from rather large numbers of study participants, the Fray study involves a single, relatively small cohort of animals, and so the numbers may not reflect a broader evaluation of more animals across more studies. Moreover, the Fray study involved animals that were infected with SIV for a year, at which point SIV infected macaques can begin progressing to AIDS. The study notably doesn’t provide CD4 count data, so the health of the animals isn’t clear, but the rather late stage of infection could have had an impact on the reported reservoir decay rates, resulting in larger differences in decay rates when compared with humans than might be the case for more typical NHP studies.

6.       Table 1. For many of the studies, the column “primary virological endpoint” doesn’t show the key virological endpoint for the actual studies.  Many of the listed studies were primary interested in evident viral reactivation (PVL increases or cell-associated viral RNA increases).  Recommend changing this column heading to “Reservoir measurement assessed”

7.       Figure 1. The figure legend should explain what the “NCT” numbers shown on the figure mean.  One of the NCT arrows seems to point the SIV line, which could lead to some confusion.

8.       Figure 2. This figure is quite similar to one that is in Bender et al. Not sure if there may be copyright issues to consider.

9.       While reading the manuscript, I got the distinct sense that the writing process was rushed and/or that the draft wasn’t carefully proofread, leading to a number of minor factual errors or mix ups, typos/misspellings or grammatical issues, and some citation issues. I note those that I readily identified while reviewed the manuscript, but I strongly recommend the author carefully read it again and double check all of the details and references list for accuracy.

a.       Line 9, Abstract: The word “are” should be “is”

b.       Line 27: the first word “Antiretroviral” is misspelled

c.       Line 35: “Progressive nonhuman primate (NHP) hosts…” This phrasing is awkward. Recommend changing the sentence to “Nonhuman primate (NHP) hosts of progressive simian immunodeficiency virus (SIV) infection, namely….”

d.       Line 58: the word “infected” is missing

e.       Line 74: Change “have shown” to “showed”

f.        Table 1: Policicchio, Borducchi, and Romidepsin are all misspelled. G-9620 should “GS-9620.” The virus used in Del Prete et al was SIVmac239X.

g.       Lines 185-186: “Fray et al (65, 66).” Reference 66 is a Kumar et al study, and reference 65 appears to be a incorrect reference altogether.

h.       Line 264: “present” should be “presents”

i.         Line 289: Presumably the number in parentheses should be a citation number

j.         Line 290: Is reference #2 correct here?

k.       Line 358: The study in question used SIVmac239, not SIVmac251

l.         Lines 523-526: A citation is needed

Minor Points:

Lines 10-11: “The utility of a model ultimately rests on how accurately it can inform therapeutic approaches” – this is something of a controversial statement and I’m not sure that all would agree.

Lines 11-13: “…and while reservoirs in the NHP model behave quantitatively very similar(ly) to those of long-term suppressed persons living with HIV-1…” I suppose it isn’t clear to me what the author means by “behave quantitatively”, but the rest of the article seems to lay out quite the opposite – that the quantitative aspects of reservoirs in NHPs are quite different from most humans on ART (who started treatment later and were treated for longer)

Line 29-30“The source of viral rebound is a pool of anatomically dispersed resting memory CD4 T cells that harbor latent but inducible HIV-1 DNA.” I’m not aware of any study that has shown that viral rebound originates from resting memory CD4 T cells.  This sentence requires a citation to support it or it should amended for accuracy.

Comments on the Quality of English Language

Overall, the manuscript could use another careful readthrough for editing and corrections.

Author Response

Major Points

  1. The weakest section of the manuscript is the “Timing of ART initiation in SIV-infected macaques” section. The author endeavors to make some generalizable points about the potential impact of ART timing on reservoir and cure studies, but many of the points seem to ultimately contradict one another and the message gets quite muddled. Initially the author spends some time nicely laying out the dynamic shift in cellular phenotypes harboring viral genomes that occurs during acute infection, identifying “0-8 weeks” (line 90) as the time window when this evolution in reservoir cell type occurs. The author then notes that a potential shortcoming of many SIV studies is the common use of ART initiation during this dynamic time period. However, the author then states that one of the justifications used by SIV researchers start ART during these early time periods is because “Initiating ART shortly after reservoir set point thus allows HIV cure approaches to be evaluated on fully matured reservoirs” (line 97). It is thus unclear when the author suggests reservoirs are still dynamically changing vs fully mature.  This point seems to suggest that even though SIV researchers are starting ART early, they are still actually past the point when the reservoir cellular phenotypes are dynamically changing, which runs counter to the original point the author was making.  This section could use some revision for clarity, as the point gets lost or muddled.

I thank the reviewer for their helpful comments.  The reviewer cogently points out that use of the term ‘fully matured’ is rather ambiguous here.  During the very dynamic window of acute HIV/SIV infection, the reservoir may reach maturity by one metric (ie, size), but still evolving in regards to others (diversity, maturation subtype composition).  I agree that omission of this nuance in the first draft leads to confusion in time and context of ART initiation.  To address this point, the most siginficant revision I have made is to clearly demarcate the window of 1-8 weeks of infection by Fiebig stage and how size, diversity, and cell type distribution of the reservoir change across this period (LINES 103-141).  Equally important, I have included a new illustration which demarcates the Fiebig stages and overlays these on ART initiation periods of various NHP studies that are discussed in the ‘ART timing’ section (Figure 1A).  In panel 1B, qualitative metricts (cell type distribution) are superimposed to illustrate to the reader the relative ‘maturity’ of each metric at the particular Fiebig stage.  We hope that this help illustrate for the reviewer the very dynamic window of reservoir evolution during these typical ART initiation windows employed by many NHP studies (weeks 1-8 post SIV).

  1. Related to point 1, line 105 states that “Very few cure approaches have been evaluated under small, less mature reservoirs that are associate with very early ART.” The author then seems to try to make a broad point thdoes not define what “very early ART” means, but the prior passages would suggest that the author means starting ART <8 weeks post infection.  In the next sentence, the author states “Ones that have however were found to elicit favorable impact and in some instances promoted sustained ART-free remission.”  However, Table 1 shows a number of studies where most initiated ART during this early period when the cellular phenotypic composition of the reservoir is still changing dynamically, but the outcomes were a mixed bag with most studies showing no evident impact on the reservoir.  Even if the author is actually referring to ART initiated during an even earlier window than this, say <2 weeks post-infection, the suggestion that earlier ART increases the likelihood of a promising result doesn’t seem consistent with the field as a whole. The vestolimod example given in the paper itself actually seems to make the counterpoint, where a study involving earlier ART showed no effect but a study involving later ART showed a dramatic effect.  I think a much safer and accurate position would be that the timing of ART initiation can impact reservoir composition, immune responses, and numerous other factors which may impact the activity (or the ability to measure activity) in ways that are intervention-specific.

I agree with the author and hope that demarcating reservoir evolution more clearly by Fiebig stage in the text and providing the acomanying Figure 1 will resolve this criticism.  By ‘early ART’, it was meant prior to establishment of set point in reservoir size (FI-III).‘later ART’ was considered after set point (Fiebgi IV-VI).  In the revised text I have removed these ambiguous terms and noted specifically the fiebig stage.  The reviewer cogently points out that many studies cited in Table 1 turned out to be variable in efficacy.  We more accurately describe these studies in lines 228-316 and note the particular fiebig stage.  The reviewer also cogently points out that the del prete study appears to, somewhat paradoxically, go contrary to the idea of earlier ART begetting greater therapeutic efficacy. I include this point  in lines 292-296 and note that the variable of ART timing on endpoint measurements in cure studies may be specific to the intervention.  Finally, I thank the reviewer for guidance in the overall summary statement of this section, provided in lines 296-317, which I believe is a more nuanced and accurate position.

  1. Line 132: “To date, therapies tested on fully-formed reservoirs have yet to reduce the replication competent viral DNA to a degree that results in a meaningful delay in viral rebound.” Related to the above 2 points, the Lim et al vesatolimod study would seem to run counter to this point.

In the revised I have noted and discus the lim study specifically (lines 292-296) noting that this particular study is generally an exception to others of similar ART initiation periods.

  1. Much is made about the potential impact of unintegrated forms of DNA, such as 2-LTR circles, on viral DNA measurements based on the fact that the levels of unintegrated DNA are higher in SIV infected macaques on ART that in humans on ART with HIV. I think this point is a bit overblown.  Although it is true that the relative levels are higher in monkeys, they still represent a very small fraction of the total DNA population during ART, so potential changes in this pool seem unlikely to meaningfully impact total DNA levels in a way that would lead to data misinterpretation.

In lines 525-531 that discuss 2-LTR circles in the ‘decay kinetics’ section I would prefer to include this point, if amendable to the reviewer, as I feel it is still important to discuss that 2-LTR fractions may comprise a greater fraction of the reservoir during interventional windows in NHP versus clinical studies.  In conlusions discussing study guidelines however, I have removed this as a ‘guideline’, as I agree that the point is too forward.  Historically many rigorous NHP studies have been performed without 2-LTR correction and with the IPDA being an exception, 2-LTR correction is conventionally not performed in other viral quantitation assays.  I thank the reviewer for reigning this in.

  1. Because this review article is focused the nuanced issues surrounding comparisons between reservoir assessments in SIV infected macaques and humans with HIV, I think that the article would benefit from a bit more nuanced discussion of the Fray et al study, rather than viewing the Fray et al numbers as necessarily indicative of decay rates across SIV/NHP studies.  In particular, while reservoir decay rates in humans have typically been derived from rather large numbers of study participants, the Fray study involves a single, relatively small cohort of animals, and so the numbers may not reflect a broader evaluation of more animals across more studies. Moreover, the Fray study involved animals that were infected with SIV for a year, at which point SIV infected macaques can begin progressing to AIDS. The study notably doesn’t provide CD4 count data, so the health of the animals isn’t clear, but the rather late stage of infection could have had an impact on the reported reservoir decay rates, resulting in larger differences in decay rates when compared with humans than might be the case for more typical NHP studies.

I am extremely thankful to the reviewer for providing this perspective, which I did not think about.  We have included these caveats in lines 431-439.

  1. Table 1. For many of the studies, the column “primary virological endpoint” doesn’t show the key virological endpoint for the actual studies.  Many of the listed studies were primary interested in evident viral reactivation (PVL increases or cell-associated viral RNA increases).  Recommend changing this column heading to “Reservoir measurement assessed”

This has been revised in column heading in Table 1.

  1. Figure 1. The figure legend should explain what the “NCT” numbers shown on the figure mean.  One of the NCT arrows seems to point the SIV line, which could lead to some confusion.

This has been revised in now Figure 2.

  1. Figure 2. This figure is quite similar to one that is in Bender et al. Not sure if there may be copyright issues to consider.

I do agree that this figure is very similar however I do not know of a more clear way to depict intactness in a figure that incorporates the IPDA.  I have inserted ‘adapted by bender et al’ with corresponding PMID to hopefully address this point.

  1. While reading the manuscript, I got the distinct sense that the writing process was rushed and/or that the draft wasn’t carefully proofread, leading to a number of minor factual errors or mix ups, typos/misspellings or grammatical issues, and some citation issues. I note those that I readily identified while reviewed the manuscript, but I strongly recommend the author carefully read it again and double check all of the details and references list for accuracy.
  2. Line 9, Abstract: The word “are” should be “is”

This has been amended.

  1. Line 27: the first word “Antiretroviral” is misspelled

This has been amended.

  1. Line 35: “Progressive nonhuman primate (NHP) hosts…” This phrasing is awkward. Recommend changing the sentence to “Nonhuman primate (NHP) hosts of progressive simian immunodeficiency virus (SIV) infection, namely….”

This has been amended.

  1. Line 58: the word “infected” is missing

This has been amended.

  1. Line 74: Change “have shown” to “showed”

This has been amended.

  1. Table 1: Policicchio, Borducchi, and Romidepsin are all misspelled. G-9620 should “GS-9620.” The virus used in Del Prete et al was SIVmac239X.

These have been amended.

  1. Lines 185-186: “Fray et al (65, 66).” Reference 66 is a Kumar et al study, and reference 65 appears to be a incorrect reference altogether.

These have been amended to the correct citations.

  1. Line 264: “present” should be “presents”

This has been amended.

  1. Line 289: Presumably the number in parentheses should be a citation number

This has been amended.

  1. Line 290: Is reference #2 correct here?

This is correct to a degree, as viremia does rebound in macaques with ART interruption.  In this sentence we speak only of HIV-1, and we have amended for accuracy

  1. Line 358: The study in question used SIVmac239, not SIVmac251

This has been amended.

  1. Lines 523-526: A citation is needed

 This has been ammended

Minor Points:

Lines 10-11: “The utility of a model ultimately rests on how accurately it can inform therapeutic approaches” – this is something of a controversial statement and I’m not sure that all would agree.

We have revised this statement to instead read “the utility of a model ultimately rests on how accurately it can recapitulate human disease.”

Lines 11-13: “…and while reservoirs in the NHP model behave quantitatively very similar(ly) to those of long-term suppressed persons living with HIV-1…” I suppose it isn’t clear to me what the author means by “behave quantitatively”, but the rest of the article seems to lay out quite the opposite – that the quantitative aspects of reservoirs in NHPs are quite different from most humans on ART (who started treatment later and were treated for longer)

On advice of the reviewer we have added the phrase quantitative differences by the “most salient aspects”, which we hope will illustrate that very-top level dynamics of the reservoir in NHP, such as viral reservoir decay in NHPs with ART (regardless of the rate) are similar despite potentially not-insignificant nuances we discuss throughout the review.

Line 29-30“The source of viral rebound is a pool of anatomically dispersed resting memory CD4 T cells that harbor latent but inducible HIV-1 DNA.” I’m not aware of any study that has shown that viral rebound originates from resting memory CD4 T cells.  This sentence requires a citation to support it or it should amended for accuracy.

We have added references to support the source of viral rebound as CD4 T cells and removed ‘resting’.

Reviewer 2 Report

Comments and Suggestions for Authors

Summary: This review discusses the use of non-human primate models to evaluate therapeutic approaches to cure HIV-1 and the similarities and differences relative to human clinical trials. While viral reservoir in NHP models resemble that of long-term suppressed people living with HIV-1 (PLWH) in many ways, important differences can impact how cure strategies evaluated in nonhuman primates translate into humans. Nuances related to the intrinsic biology of both the virus and host, as well as the timeframe of the disease course assessed in NHP compared with PLWH are highlighted. In particular, the higher decay rate after ART, the proviral intactness of SIV reservoirs and its less clonal structure are notable variations.  Major emphasis is placed on differences in the timing of ART initiation between NHP studies and clinical trials, and the impact this variable has on the decay and clonality of reservoirs. Interpretation of cure interventions in NHP, including cases of discordant outcomes, and how they might be expected to predict outcomes in humans is discussed. Manuscript strengths are that it is clear and well-written, and adeptly weaves together the literature on NHP and human reservoir analyses and therapeutic interventions.

General concept comments:

The review offers some general guidelines to NHP investigators regarding study design in the Concluding Remarks with respect to when ART should be initiated, suggesting that the most clinically relevant timing would be > 4 weeks post-infection.  However, in the field of HIV therapeutics, interventions yielding any efficacy signal in rigorous NHP studies are limited.  Allowing infection to proceed to >4 weeks raises the bar by permitting further establishment of reservoirs through a longer period of viral replication.  Consideration of this challenge is warranted, as most NHP investigators will prioritize an efficacy signal as a proof of concept for their test intervention over a design that is “most clinically relevant.”  Similarly, human clinical trials evaluating HIV-1 therapeutics often seek to enroll people in the earliest stages of infection because it offers a potentially lower bar for cure.

Specific comments:

1)    Line 42: NHP do not represent a single model.  Using different NHP species or viruses all represent different models.  Recommend, “NHP models can accurately…”

2)    Line 77: RV421 cohort is incorrect.  The author is likely referring to RV217 (ECHO; PMID: 27192360).  Another relevant cohort not mentioned is the FRESH cohort from S. Africa.  To this reviewer’s knowledge, reservoir analyses on the MERLIN cohort from Peru have not been as extensive and perhaps should not be included in this list.  Citations referencing the pertinent cohorts should be included.  Also, RV254 does not enroll people at high risk of acquiring HIV, as stated.  Rather, it identifies individuals at the time of anonymous HIV test results indicating viremia in the absence of HIV-1 antibody responses.

3)    Additional references with probable errors:

a.     Table 1: Lim et al: ref is #12 (not #7)

b.     Line 116: Confirm that ref# 33 illustrates the variability of SHIV-SF162P3 replication and pathogenicity.

c.     Line 179&185: Confirm that ref #65 refers to the intended article

d.     Lines 186 through 206- the study cited in ref#66 uses SHIV genomes and not SIV. It looks like the correct reference would be PMID: 36809762 instead. It appears correctly cited lines 230-233

e.     Line 289: reference not formatted?

4)     Line 183: Using the term, “viral genomes” to refer to viral DNA that is either integrated or not during the early phases of decay instead of the term “proviruses” is confusing since the HIV/SIV genome is RNA, not DNA.  “Total DNA” is commonly used in the literature,

5)     Line 205 is missing a reference supporting the 4th phase of HIV reservoir dynamics

6)     Figure 1B would benefit from overlaying human/HIV comparators, as in A

7)     Line 347 should include citations for the 2 references being discussed.

8)     Line 402: “specific” pathogen-free rhesus, not “specialized”

Comments on the Quality of English Language

Overall, English quality is very good.  Several instances of capitalization (e.g. persons (line27), elite controllers, simian immunodeficiency virus) and italicization (e.g. viral gene names, such as env) require attention.

Author Response

Summary: This review discusses the use of non-human primate models to evaluate therapeutic approaches to cure HIV-1 and the similarities and differences relative to human clinical trials. While viral reservoir in NHP models resemble that of long-term suppressed people living with HIV-1 (PLWH) in many ways, important differences can impact how cure strategies evaluated in nonhuman primates translate into humans. Nuances related to the intrinsic biology of both the virus and host, as well as the timeframe of the disease course assessed in NHP compared with PLWH are highlighted. In particular, the higher decay rate after ART, the proviral intactness of SIV reservoirs and its less clonal structure are notable variations.  Major emphasis is placed on differences in the timing of ART initiation between NHP studies and clinical trials, and the impact this variable has on the decay and clonality of reservoirs. Interpretation of cure interventions in NHP, including cases of discordant outcomes, and how they might be expected to predict outcomes in humans is discussed. Manuscript strengths are that it is clear and well-written, and adeptly weaves together the literature on NHP and human reservoir analyses and therapeutic interventions.

General concept comments:

The review offers some general guidelines to NHP investigators regarding study design in the Concluding Remarks with respect to when ART should be initiated, suggesting that the most clinically relevant timing would be > 4 weeks post-infection.  However, in the field of HIV therapeutics, interventions yielding any efficacy signal in rigorous NHP studies are limited.  Allowing infection to proceed to >4 weeks raises the bar by permitting further establishment of reservoirs through a longer period of viral replication.  Consideration of this challenge is warranted, as most NHP investigators will prioritize an efficacy signal as a proof of concept for their test intervention over a design that is “most clinically relevant.”  Similarly, human clinical trials evaluating HIV-1 therapeutics often seek to enroll people in the earliest stages of infection because it offers a potentially lower bar for cure.

We thank the reviewer for this overall comment, and it raises a very cogent point.  We agree that timing of ART initiation should be considered within the context of each individual study, and should be weighed against the investigator’s priority of establishing proof of priniciple (earler ART) of a therapeutic approach versus its translatability (later ART).  We thus feel in the original draft that suggesting a definitive timepoint of ‘> 4 weeks’ is too rigid and would not be an appropriate guideline, as it would de-emphasize approaches that may be able to meet a therapeutic bar on less mature reservoirs which would nonetheless advance the field of cure therapeutics.  In the revised version we are thus less restrictive with the timing of ART, rather, we emphasize the relationship between ART timing to proof-of-principle versus clinical relavence (lines 143-151).

Specific comments:

  • Line 42: NHP do not represent a single model.  Using different NHP species or viruses all represent different models.  Recommend, “NHP models can accurately…”

This has been amended in the revised

  • Line 77: RV421 cohort is incorrect.  The author is likely referring to RV217 (ECHO; PMID: 27192360).  Another relevant cohort not mentioned is the FRESH cohort from S. Africa.  To this reviewer’s knowledge, reservoir analyses on the MERLIN cohort from Peru have not been as extensive and perhaps should not be included in this list.  Citations referencing the pertinent cohorts should be included.  Also, RV254 does not enroll people at high risk of acquiring HIV, as stated.  Rather, it identifies individuals at the time of anonymous HIV test results indicating viremia in the absence of HIV-1 antibody responses.

I thank the reviewer for pointing out these original oversights.  In the revised, I have more accurately noted the clinical cohorts that have interrogated the HIV-1 reservoir quantitatively and qualitatively in hyperacute HIV-1 infection (lines 97-99). We have also included a new illustration (Figure 1) that depicts these qualitative and quantitative aspects of reservoir formation across Fiebig stage.  We overlay these to ART initiation timepoints of various NHP studies shown in Table 1.

3)    Additional references with probable errors:

  1. Table 1: Lim et al: ref is #12 (not #7)

This has been ammended in the revised.

  1. Line 116: Confirm that ref# 33 illustrates the variability of SHIV-SF162P3 replication and pathogenicity.

The original is correct.

  1. Line 179&185: Confirm that ref #65 refers to the intended article

This has been amended in the revised

  1. Lines 186 through 206- the study cited in ref#66 uses SHIV genomes and not SIV. It looks like the correct reference would be PMID: 36809762 instead. It appears correctly cited lines 230-233

Several instances of this have been amended in the revised (ref 26),

  1. Line 289: reference not formatted?

This has been amended in the revised

  • Line 183: Using the term, “viral genomes” to refer to viral DNA that is either integrated or not during the early phases of decay instead of the term “proviruses” is confusing since the HIV/SIV genome is RNA, not DNA.  “Total DNA” is commonly used in the literature,

In most instances I have substituted ‘viral genomes’ to ‘viral DNA’.  In some instances I have kept the term ‘viral genomes’ to be consistent with verbiage noted in the referenced study, particularly that of Fray et al.

  • Line 205 is missing a reference supporting the 4thphase of HIV reservoir dynamics

This has been included in the revised.

  • Figure 1B would benefit from overlaying human/HIV comparators, as in A

In the original I have drawn down the shaded blue window of 1A to encompass the clonal landscape and 2-LTR circles in 1B.  In the revised version I have also added the same clinical trial numbers to figure 1B as shown in

  • Line 347 should include citations for the 2 references being discussed.

This has been amended.

  • Line 402: “specific” pathogen-free rhesus, not “specialized”.

This has been ammended in the revised.

Comments on the Quality of English Language

Overall, English quality is very good.  Several instances of capitalization (e.g. persons (line27), elite controllers, simian immunodeficiency virus) and italicization (e.g. viral gene names, such as env) require attention.

These have been ammended in the revised.

Reviewer 3 Report

Comments and Suggestions for Authors

This is a well written review, Mudd does a thorough job of covering the pros and cons of using NHP models to study the HIV reservoir. The conclusions drawn and recommendations presented will be valuable to the field and hopefully result in researchers attempting to answer these questions using similarly designed studies. I have a few relatively minor comments/ questions for the author to consider.

1.        I have been told several times now that the field is moving away from PLWH and that people with HIV (PWH) is now the preferred term.

2.        Figure 1B is rather confusing. The text suggests that both NHP and clinical data will be presented but it is not clear what is being presented in this figure. 1A does a nice job of separating the SIV and HIV data as well as mentioning the studies it is referring to (for clinical).  Additionally, 1B has no timeline on the x axis.  

3.        Figure 2 - in the box with the NHP differences in IPDA, it may be worth adding that total genomes cannot be reported. This is stated nicely in the text but is worth highlighting in the figure as well.

4.        Clonotypic structure of the reservoir – The discussion on why NHPs have less clonal provirus is interesting but the point about the animals having similar exposure as people to recurrent antigens too strong. Though time is likely the largest contributor to lack of clonality in NHP reservoirs, environmental exposure likely also plays a role. As NHP in facilities lack ongoing immune exposure even if they are CMV positive when entering study. Therefore, this may play a larger role than originally thought. The author alludes to this with the proposed CMV experiment later in the section.

5.        Conclusions – The recommendation made for future NHP studies are excellent. This reviewer would suggest adding a qualifier to ART treatment time and suggest a minimum of 36 weeks of ART when the study design allows. The Siliciano group nicely showed in Bender et al that the CD4 reservoir starts to plateau around 36 weeks in animals that were treated with ART after 95 weeks of viremic infection, which is substantially longer than most studies can afford, so one can assume short viremic periods may plateau similarly.

6.        Conclusions – line 514 - Large proviral clones have also been shown in elite controllers despite sustain virological control without ART (30232278)

7.        Conclusions – This reviewer appreciates the mention of other HIV/SIV reservoirs. However, it should be noted that in NHPs myeloid reservoirs have been detect across most of the major strains used to date SIVmac239 (37112989), SIVmac251(you have cited #126), and B670/Fr (31118264, 31789817). Though, I agree that specific strains are related to the development of severe CNS disease, macrophage reservoirs in general are not likely a result of a specific strain.  

8.        General question – Is there any data in Africa green monkeys that could shed light on the generic diversity of the provirus? Ie why NHP tend to have such high levels of APOBEC mutations. Could it be a result of introducing a virus into a host that has not evolved a long side it?

Author Response

Reviewer 3

Comments and Suggestions for Authors

This is a well written review, Mudd does a thorough job of covering the pros and cons of using NHP models to study the HIV reservoir. The conclusions drawn and recommendations presented will be valuable to the field and hopefully result in researchers attempting to answer these questions using similarly designed studies. I have a few relatively minor comments/ questions for the author to consider.

  1. I have been told several times now that the field is moving away from PLWH and that people with HIV (PWH) is now the preferred term.

I thank the reviewer for their helpful comments, the ‘PLWH’ acronym has been ammended in all instances to ‘PWH’ in the revised.

  1. Figure 1Bis rather confusing. The text suggests that both NHP and clinical data will be presented but it is not clear what is being presented in this figure. 1A does a nice job of separating the SIV and HIV data as well as mentioning the studies it is referring to (for clinical).  Additionally, 1B has no timeline on the x axis.

For clarity in the revised (now figure 2), I have added a timeline to the same scale as panel A.  The blue shaded portion of the ART timeline is dragged down to encompass both panel A and B

  1. Figure 2 - in the box with the NHP differences in IPDA, it may be worth adding that total genomes cannot be reported. This is stated nicely in the text but is worth highlighting in the figure as well.

This has been added in the revised, now figure 3.

  1. Clonotypic structure of the reservoir– The discussion on why NHPs have less clonal provirus is interesting but the point about the animals having similar exposure as people to recurrent antigens too strong. Though time is likely the largest contributor to lack of clonality in NHP reservoirs, environmental exposure likely also plays a role. As NHP in facilities lack ongoing immune exposure even if they are CMV positive when entering study. Therefore, this may play a larger role than originally thought. The author alludes to this with the proposed CMV experiment later in the section.

We do agree with the reviewer that an underappreciated aspect of clonal expansion could be related to environmental antigens, which may differ among macaque colonies at various research primate centers depending upon the housing facility (indoor vs outdoor).  In regards to the specific sentence the reviewer highlights, in the revised I have clarified more narrowly the form of recurring antigens we are referring to (ie, persistent beta-herpesviral infections)(Line 680).  I hope this will be more suitable to the reviewer.

  1. Conclusions– The recommendation made for future NHP studies are excellent. This reviewer would suggest adding a qualifier to ART treatment time and suggest a minimum of 36 weeks of ART when the study design allows. The Siliciano group nicely showed in Bender et al that the CD4 reservoir starts to plateau around 36 weeks in animals that were treated with ART after 95 weeks of viremic infection, which is substantially longer than most studies can afford, so one can assume short viremic periods may plateau similarly.

Please see revisions related to the ‘ART timing’ section we have incorporated in response to comments of the additional referees, which I hope will be more suitable to the reviewer.  In general, we have revised this section to suggest guidance on ART timing that is less restrictive to a particular timepoint, and instead have noted the potential benefits of earlier ART (proof-of-concept) versus later ART (translatability) in evaluating a particular HIV cure therapy.  In summary, ART timing will depend uniquely upon the context of each study.

  1. Conclusions – line 514 - Large proviral clones have also been shown in elite controllers despite sustain virological control without ART (30232278)

The author is well aware of a similar proviral landscape in elite controllers who have also undergone sustained long-term negative immune selection of transcriptionally-active proviruses.  The purpose of this sentence however is to compare landscapes in the setting of long-term ART in PWH (larger window to facilitate immune selection) versus the relatively condensed ART timeframes of NHP studies (smaller window of immune selection).  Because elite controllers represent a different context in this regard (generally ART-naïve), I prefer to not include the mention of this population so as to not confuse the reader, and hope it would be amendable to the reviewer to leave as is.

  1. Conclusions– This reviewer appreciates the mention of other HIV/SIV reservoirs. However, it should be noted that in NHPs myeloid reservoirs have been detect across most of the major strains used to date SIVmac239 (37112989), SIVmac251(you have cited #126), and B670/Fr (31118264, 31789817). Though, I agree that specific strains are related to the development of severe CNS disease, macrophage reservoirs in general are not likely a result of a specific strain.

The reviewer raises an excellent point, it is more accurate that  macrophage tropism of SIV strains can most saliently impact their pathogenicity in the CNS.  But indeed, viral DNA has been observed in myeloid cells regardless of the SIV strain employed.  We have removed this sentence as it was helpfully pointed out by the reviewer that it is not entirely accurate, and have additionally added the references above for SIVmac239 and B670/Fr strains (Line 835).

  1. General question – Is there any data in Africa green monkeys that could shed light on the generic diversity of the provirus? Ie why NHP tend to have such high levels of APOBEC mutations. Could it be a result of introducing a virus into a host that has not evolved a long side it?

To my knowledge I do not know of any studies that have interrogaged viral diversity by env or whole genome in natural hosts of SIV such as AGMs.  I would imagine that viral diversity is high in AGM and continually evolving given the ongoing replication in natural hosts, despite infection remaining asymptomatic.  The higher levels of apobec mutations noted in progressive hosts relative to humans are indeed interesting.  Either viral or host factors could underlie this difference although I do not know of reports examine this phenomenon. 

Round 2

Reviewer 1 Report

Comments and Suggestions for Authors

The revised review is very nice, with clear, well-supported points and an appropriate amount of detail.